# Formulation Strategies to Improve Nose-to-Brain Delivery of Donepezil

**DOI:** 10.3390/pharmaceutics11020064

**Published:** 2019-02-01

**Authors:** Lupe Carolina Espinoza, Marcelle Silva-Abreu, Beatriz Clares, María José Rodríguez-Lagunas, Lyda Halbaut, María-Alexandra Cañas, Ana Cristina Calpena

**Affiliations:** 1Department of Pharmacy, Pharmaceutical Technology and Physical Chemistry, Faculty of Pharmacy and Food Sciences, University of Barcelona, 08028 Barcelona, Spain; lcespinoza@utpl.edu.ec (L.C.E.); marcellesabreu@gmail.com (M.S.-A.); halbaut@ub.edu (L.H.); 2Departamento de Química y Ciencias Exactas, Universidad Técnica Particular de Loja, Loja 1101608, Ecuador; 3Institute of Nanoscience and Nanotechnology (IN2UB), University of Barcelona, 08028 Barcelona, Spain; beatrizclares@ugr.es; 4Pharmacy and Pharmaceutical Technology Department, Faculty of Pharmacy, University of Granada, 18071 Granada, Spain; 5Department of Biochemistry and Physiology, Faculty of Pharmacy and Food Sciences, University of Barcelona, 08028 Barcelona, Spain; mjrodriguez@ub.edu (M.J.R.-L.); marialexandra5@gmail.com (M.-A.C.); 6Institut de Recerca en Nutrició i Seguretat Alimentària (INSA), Universitat de Barcelona (UB), 08028 Barcelona, Spain; 7Biosanitary Institute of Granada (ibs.GRANADA), University Hospitals of Granada-University of Granada, 18012 Granada, Spain

**Keywords:** Donepezil, Alzheimer’s disease, nanoemulsion, mucoadhesion, nose-to-brain, Pluronic F-127

## Abstract

Donepezil (DPZ) is widely used in the treatment of Alzheimer’s disease in tablet form for oral administration. The pharmacological efficacy of this drug can be enhanced by the use of intranasal administration because this route makes bypassing the blood–brain barrier (BBB) possible. The aim of this study was to develop a nanoemulsion (NE) as well as a nanoemulsion with a combination of bioadhesion and penetration enhancing properties (PNE) in order to facilitate the transport of DPZ from nose-to-brain. Composition of NE was established using three pseudo-ternary diagrams and PNE was developed by incorporating Pluronic F-127 to the aqueous phase. Parameters such as physical properties, stability, in vitro release profile, and ex vivo permeation were determined for both formulations. The tolerability was evaluated by in vitro and in vivo models. DPZ-NE and DPZ-PNE were transparent, monophasic, homogeneous, and physically stable with droplets of nanometric size and spherical shape. DPZ-NE showed Newtonian behavior whereas a shear thinning (pseudoplastic) behavior was observed for DPZ-PNE. The release profile of both formulations followed a hyperbolic kinetic. The permeation and prediction parameters were significantly higher for DPZ-PNE, suggesting the use of polymers to be an effective strategy to improve the bioadhesion and penetration of the drug through nasal mucosa, which consequently increase its bioavailability.

## 1. Introduction

Alzheimer’s disease (AD) is a neurodegenerative disease characterized by progressive cognitive dysfunction, memory loss, and difficulties carrying out daily routine activities [1]. The exact etiology of AD is still unknown although it is considered a multifactorial disease whose pathogenesis includes neuro-cholinergic changes as well as development of amyloid plaques and neurofibrillary tangles [2,3,4]. To date, there are four pharmacological options approved for the symptomatic treatment of AD: Memantine, which is a *N*-methyl-d-aspartate (NMDA) receptor antagonist, and three acetylcholinesterase (AChE) inhibitors, namely Rivastigmine, Donepezil (DPZ), and Galantamine [5]. DPZ is a non-competitive and reversible AChE inhibitor with high selectivity in both targets and tissues. Its main action is to increase cholinergic transmission, although other mechanisms of DPZ for AD treatment have been reported, including a decrease of neural toxicity caused by β-amyloid protein, prevention of the reduction in nicotinic binding, and decline of glutamate-induced neurotoxicity [6]. These pharmacological activities enable this drug to have significant efficacy and as a result it is widely used to reduce the severity of neuropsychiatric symptoms as well as to improve the cognitive capacity of patients with mild-to-moderate AD [7]. DPZ is commercially available in solid dosage forms (5, 10, and 23 mg) for oral administration [8]. However, this route of administration presents notable disadvantages, such as first-pass metabolism, adverse effects in the gastrointestinal system, and low bioavailability in the brain due to the drug’s poor ability to penetrate the blood–brain barrier (BBB), all of which evidence the need to use alternative routes of administration and to develop new drug delivery systems that improve pharmacological efficacy in the treatment of AD [9]. 

The intranasal route represents an alternative and painless method to noninvasively bypass the BBB [10]. The advantages of intranasal administration are manifold. For one, due to rapid absorption without enzymatic degradation and first-pass metabolism, bioavailability is improved and therefore allows rapid onset of the pharmacological action. Additionally, reduced systemic exposure minimizes drug distribution to non-targeted sites, thus reducing adverse side effects. Furthermore, this route allows the delivery of drug directly to the brain from the nasal cavity via olfactory and trigeminal nerve pathways [11,12,13]. This route is also convenient for geriatric patients unable to swallow oral dosage forms [14]. Despite these benefits, there are also some disadvantages associated with intranasal route administration, such as low epithelial permeability, limited surface area, enzymatic degradation, small volume used for drug administration, and mucociliary clearance that leads to the removal of drugs from the site of absorption [15]. Therefore, formulation strategies that ought to be adapted in order to achieve the success of the treatment using the nasal route include improvement of solubility and permeability, extension of residence time, and protection against enzymatic degradation [9,16,17]. 

The use of nanocarriers as a promising tool to overcome the limitations of the intranasal route has been gaining interest over recent decades. Nanotechnology-based drug delivery systems (polymeric nanoparticles, solid lipid nanoparticles, nanostructured lipid carriers, microemulsions, nanoemulsions, and liposomes) are characterized by their nanoscale size range, which can be appropriate to transport drugs to the brain and enhance the therapeutic performance of active agents used in the treatment of neurodegenerative disorders [15,18]. 

Nanoemulsions (NEs) are colloidal nanocarriers of low viscosity formed by droplets with size in nanometer scale usually <250 nm and characterized by being kinetically stable with a transparent and translucent appearance with a bluish tint due to their weak interaction with light [15,19,20]. NEs are promising systems for intranasal delivery due to their reduced droplet size, high permeability, solubilizing potential, and, in particular, their protecting effect for lipophilic molecules [21]. However, one drawback of this type of formulation is its rapid nasal clearance, but this can be readily overcome by the addition of mucoadhesive and mucus-penetrating agents in order to increase the residence time at the site of absorption and improve the penetration of drug through the mucosa layer, which subsequently enhances bioavailability [22,23]. 

Considering these remarkable findings, the purpose of this study was to develop and characterize a nanoemulsion (NE), as well as a nanoemulsion containing Pluronic F-127 (PNE) as polymer, to improve the bioadhesion and penetration of drug in order to facilitate the transport of DPZ from nose-to-brain and thereby increase its efficacy in the treatment of AD.

## 2. Materials and Methods 

### 2.1. Materials

The DPZ was purchased from Capot Chemical (Hangzhou, China). Plurol oleique CC 497 (polyglyceryl-3 dioleate), Labrafil M1944 CS (oleoyl polyoxyl-6 glycerides), Labrafac lipophile WL 1349 (medium chain triglycerides), Capryol 90 (propylene glycol monocaprylate), Lauroglycol 90 (propylene glycol monolaurate), Labrasol (caprylocaproyl polyoxyl-8 glycerides), and Transcutol-P (diethylene glycol monoethyl ether) were all supplied by Gattefossé (Saint-Priest, France). Macrogolglycerol ricinoleate (Cremophor EL) was obtained from Fagron Ibérica (Barcelona, Spain). Tween 80, castor oil, polyethylene glycol, and propylene glycol were purchased from Sigma-Aldrich (Madrid, Spain). Mucin from porcine stomach was purchased from Sigma Aldrich (Madrid, Spain). Components for histological assays were acquired from Sigma and Thermo Fisher Scientific (Barcelona, Spain). Chemicals for analytical experiments were purchased from Panreac (Barcelona, Spain) and the water used was obtained from a Millipore Milli-Q purification system (Millipore Corporation, Burlington, MA, USA).

### 2.2. High-Performance Liquid Chromatography (HPLC) 

DPZ was determined using a Waters HPLC with 2487 (UV/Vis) Detector & 717 Plus Autosampler (Waters, Milford, MA, USA). The assay was carried out using a Kromasil C_18_ column (250 × 4.6 mm × 5 µm). The mobile phase was a mixture of methanol:buffer (50:50, *v*/*v*) which was filtered using a 0.45 µm polyvinylidene fluoride (PVDF) filter (Millipore Corp., Madrid, Spain). The buffer consisted of a mixture of potassium dihydrogen orthophosphate 0.05 M, water, and trimethylamine, and by adjusting pH to 2.5 ± 0.05 with orthophosphoric acid. The mobile phase was pumped through the C_18_ column at a flow rate of 1.2 mL/min. A volume of 20 μL was injected and the elute was analyzed at 268 nm. Data were evaluated using Empower 3 software—Build 3471 (Waters, Milford, MA, USA, 2010) [24].

### 2.3. Solubility Study

The solubility of DPZ was evaluated in several oils (Capryol 90, castor oil, Plurol oleique CC 497, Labrafac lipophile WL 1349, Labrafil M1944 CS, and Lauroglycol 90), surfactants (Tween 80, Cremophor and Labrasol), and cosurfactants (Transcutol-P, propylene glycol, and polyethylene glycol). An excess of DPZ was added to 3 g of each of these oils, surfactants, and cosurfactants and then subsequently mixed at 25 °C for 6 h under stirring. The samples were equilibrated overnight at room temperature and later centrifuged at 9000 rpm for 15 min. The supernatant was diluted with a solution of methanol:water (50:50, *v*/*v*). DPZ was subsequently determined at 312 nm using a DR 6000 UV-Visible Spectrophotometer (Hach, Düsseldorf, Germany).

### 2.4. Construction of Pseudo-Ternary Phase Diagrams

Three pseudo-ternary phase diagrams were constructed by water titration method using the components that exhibited the highest solubilizing potential for DPZ. Capryol 90 was selected as the oil phase, a mixture (S_mix_) of Labrasol as surfactant, Transcutol-P as cosurfactant at three different ratios (1:1, 2:1 and 3:1), and purified water as the aqueous phase. For each pseudo-ternary diagram, oil and S_mix_ were mixed at ratios from 9:1 to 1:9 (*w*/*w*) while the aqueous phase was added by titration until turbidity or phase separation was observable so that it was possible to delineate the boundaries of phases formed. NE area was formed only by monophasic and transparent mixtures. From the three pseudo-ternary phase diagrams obtained, the one showing the highest NE area was selected as the optimal S_mix_ ratio.

### 2.5. Preparation of DPZ-NE and DPZ-PNE

DPZ-loaded NE (6.25 mg/mL) was prepared by incorporating DPZ in oil under stirring at 700 rpm until dissolution of drug, after which S_mix_ was incorporated under the same condition of stirring. Finally, water was slowly added until obtaining transparent and homogeneous NE. To prepare DPZ-PNE, Pluronic F-127 was incorporated into the aqueous phase of NE. 

### 2.6. Characterization of DPZ-NE and DPZ-PNE

The pH of DPZ-NE and DPZ-PNE were determined using a pH meter GLP 22 (Crison Instruments, Barcelona, Spain). 

The droplet size and polydispersity index (PI) were determined by dynamic light scattering (DLS) using a Zetasizer Nano ZS (Malvern Instruments, Malvern, UK). These measurements were carried out with 1 mL of the formulation without dilution at 25 °C using polystyrene cells. Data were expressed as mean ± standard deviation (SD) of 3 replicates. 

The morphology of DPZ-NE and DPZ-PNE was examined by transmission electron microscopy (TEM) using a JEOL JEM-1010 electron microscope (JEOL Ltd., Tokyo, Japan). The sample preparation of both was carried out by negative staining with uranyl acetate and 24 h of drying. 

Entrapment efficacy (EE) was determined by centrifugation of the formulation at 5000 rpm for 50 min. The free drug present in the supernatant was measured using UV spectroscopy at 230 nm. The experiment was carried out in triplicate and EE (%) was calculated using Equation (1).
(1)$$EE\left(\%\right)=\frac{Total\;amount\;of\;DPZ-Free\;amount\;of\;DPZ}{Total\;amount\;of\;PGZ}\times 100$$

Viscosity and rheological behavior were evaluated 24 h after DPZ-NE and DPZ-PNE preparation using a Haake RheoStress 1 rheometer connected to a temperature control Thermo Haake Phoenix II + Haake C25P (Thermo Fisher Scientific, Karlsruhe, Germany) and equipped with cone-plate geometry (0.105 mm gap) including a Haake C60/2Ti mobile cone (60 mm diameter and 2° angle). Samples tested in 2 replicates at 25 °C underwent a program of 3-step shear profile: a ramp-up period (0–50 s^−1^) for 3 min, constant shear rate period at 50 s^−1^ for 1 min, and a ramp-down period (50 to 0 s^−1^) for 3 min. The data from the flow curve (shear stress (τ) versus shear rate (𝛾̇)) were fitted to the following mathematical models: Newton, Bingham, Ostwald-de-Waele, Casson, Herschel-Bulkley, and Cross. The model that best statistically describes the experimental data was selected based on the correlation coefficient value (r) and chi-squared value. The viscosity value was determined from the viscosity curve.

Mucoadhesive properties of DPZ-NE and DPZ-PNE were evaluated by falling liquid film technique [25]. Pieces of porcine nasal mucosa of 3 cm were hydrated with artificial nasal mucus prepared with 8% of mucin from porcine stomach dispersed in a solution of 7.45 mg/mL NaCl, 1.29 mg/mL, 1.29 mg/mL KCl and 0.32 mg/mL CaCl_2_·2H_2_O, and subsequently situated on a semi-cylindrical plastic tube held in an inclined position at an angle of 45°. A sample of 100 µL of each formulation was placed on 2 individual mucosal surfaces. After 15 min, 20 mL of PBS (pH 6.4) previously warmed at 37 °C were distributed by droplet flow onto the mucosa and the eluted PBS was collected in a beaker. Afterwards, the amount of DPZ collected in the eluted PBS was determined at 312 nm using a DR 6000 UV-Visible Spectrophotometer (Hach, Düsseldorf, Germany) [26,27].

### 2.7. Stability Studies

The physical stability of DPZ-NE and DPZ-PNE was studied by multiple light scattering analysis using a TurbiScanLab (Formulation, Toulouse, France). This technology has two synchronous detectors to analyze particle migration or particle size changes. The transmission detector measures the light flux transmitted through the formulation (T) while the backscattering detector measures the light backscattered by the formulation (BS). Samples of 20 mL were stored at 4, 25, and 40 °C for 45 days. Final formulations were transparent, hence only the T profile obtained after predetermined time intervals (1, 30, and 45 days) was used to study the stability of the formulation. The sample was analyzed at room temperature for 3 h and the data were obtained at intervals of 15 min.

### 2.8. In Vitro Release Study

The release study of DPZ from NE and PNE was performed using Franz diffusion cells (FDC 400; Crown Grass, Somerville, NJ, USA) and dialysis membranes (MWCO 12 KDa) previously hydrated in methanol:water (50:50, *v*/*v*) for 24 h. The effective diffusion area was 2.54 cm^2^ and the receptor volume was 13 mL. The dialysis membrane was mounted between the donor and receptor compartment. The receptor compartment was filled with receptor medium (RM) formed by methanol:Transcutol-P (50:50, *v*/*v*). The temperature and stirring rate in the system were set at 37 ± 0.5 °C and 100 rpm, respectively, to accomplish sink conditions. A sample of 0.3 mL of formulation was placed in the donor compartment. At predetermined time intervals (0.17, 0.5, 1, 2, 4, 6, 8, 10, 23, 26, and 30 h), aliquots of 0.3 mL were extracted from the receptor compartment and replaced with the same volume of fresh RM. DPZ released from NE and PNE was quantified by HPLC (Section 2.2). The experiment was carried out in triplicate and data are given as mean ± SD. The released amount of DPZ (µg) was plotted versus time (h) using GraphPad Prism^®^ 6.0 (GraphPad Software Inc., San Diego, CA, USA, 2014). Five kinetic models (first order, Hyperbolic, Higuchi, Weibull, and Korsmeyer-Peppas) were used to determine the release kinetic profile and the one with the highest *r*^2^ was subsequently selected.

### 2.9. Ex Vivo Permeation Studies

Transmucosal permeation studies were performed using porcine nasal mucosa obtained from the Animal Facility of the Faculty of Medicine, in accordance with Animal Experimentation Ethical Committee of the University of Barcelona, Spain (CEEA-UB). Sacrifice with sodium pentobarbital (250 mg/kg) was administered through the auricular vein under deep anesthesia, and afterwards the nasal mucosa membranes were removed, preserved in Hank’s balanced salt solution, and refrigerated until the initiation of the experiments. The experiment was carried out using Franz diffusion cell (FDC 400; Crown Grass, Somerville, NJ, USA) with a receptor volume of 6 mL and a diffusion area of 0.64 cm^2^. Nasal mucosa membranes with thicknesses of 500 ± 100 µm were placed between the donor and receptor compartment. A mixture of Transcutol-P:water (60:40, *v*/*v*) was used as RM and was maintained under stirring at 100 rpm. The temperature in the system was kept at 37 ± 0.5 °C. A sample of 0.3 mL of formulation was placed in the donor compartment in contact with nasal mucosa. At pre-established time intervals (0.1, 0.25, 0.4, 0.5, 0.75, 1, 2, 3, 4, and 6 h), aliquots of 0.3 mL were extracted from the receptor compartment and replaced with an equal volume of fresh RM. The determination of DPZ permeated through the nasal mucosa was performed by HPLC. Data were expressed as mean ± SD (*n* = 6). A graphical representation of the cumulative permeated amount of DPZ (µg) versus time (h) was performed and the slope of the linear stretch was determined by linear regression analysis. 

Further biopharmaceutical evaluation was performed to determine different permeation parameters, such as flux or permeation rate (*J_ss_*, µg/(min/cm^2^)), permeability coefficient (*K_p_*, (cm/min)·10^3^)). Moreover, the theoretical human steady-state plasma concentration (*C_ss_*) of drug, which estimates the concentration of drug that could be reached in the blood after nasal administration, was obtained using Equation (2).
(2)$$C_{ss}=J_{ss}\;\frac{A}{Clp}$$
where *J_ss_* is the flux, *A* is the hypothetical area of application (150 cm^2^ for nasal mucosa), and *Clp* is the plasmatic clearance (human *Clp* value for DPZ according to the Food and Drug Administration is 10 L/h) [28]. 

After permeation studies, the nasal mucosa was removed from the Franz diffusion cell and cleaned with distilled water. The DPZ retained in these samples was extracted with 1 mL of methanol using an ultrasonic bath for 20 min. The resulting solution was filtered and analyzed by HPLC to determine the amount of DPZ retained in the mucosa (*Q_ret_*, µg DPZ/g tissue/cm^2^).

### 2.10. Cytotoxicity Assay

The effect of DPZ-NE and DPZ-PNE on cell viability was evaluated using Methylthiazolyldiphenyl-tetrazolium bromide (MTT) cytotoxicity assay, which measures the reduction of tetrazolium salt realized by intracellular dehydrogenases of viable living cells. In order to perform this assay, human nasal septum carcinoma cell line RPMI 2650 (Sigma Aldrich) (2 × 10^5^ cells/mL) was plated in 96-wells plates (Corning Inc., Corning, NY, USA) and cultured in a humidified incubator at 37 °C in a 5% CO_2_ atmosphere for 24 h to facilitate adhesion [29]. Cells were grown in Eagle’s minimum essential medium (EMEM) supplemented with 2 mM glutamine, 1% non-essential amino acids (NEAA), 100 U/mL penicillin, 100 μg/mL streptomycin, and 10% heat inactivated fetal bovine serum (FBS). The cells were treated with different dilutions (3.125 to 125 µg/mL) of DPZ-NE, DPZ-PNE, and the blank formulations for 24 h, and then subsequently incubated with fresh medium and 10% MTT (5 mg/mL in phosphate buffered saline) for 2 h at 37 °C. Afterwards, the medium was removed carefully and 100 μL of dimethyl sulfoxide (DMSO) 99% purity was added to lysate the cells and dissolve the purple insoluble crystals of MTT. The cell lysate was transferred to a new 96-well plate and then the absorbance was read using a Microplate Autoreader at excitation/emission of 540/630 nm (Modulus Microplate Multimode Reader-Turner Biosystems, Sunnyvale, CA, USA). In a parallel manner, a negative control (cells without any stimulation or treatment) was processed for comparison. Absorbance values were considered directly proportional to cell viability.

### 2.11. Histological Analysis

The in vivo tolerance of the developed formulations was evaluated using pigs from the Animal Facility of the Faculty at Bellvitge Campus of the University of Barcelona in accordance with both the Animal Experimentation Ethical Committee of the University of Barcelona, Spain (CEEA-UB) and Animal Experimentation Commission of the Generalitat de Catalunya. Pigs were designated as the following groups: negative control (non-treated pig), positive control (pig treated intranasally with isopropyl alcohol), pig treated with DPZ-NE, and pig treated with DPZ-PNE. A volume of 300 µL of developed formulation or isopropyl alcohol was administered using a nasal spray pump. After 1 h of treatment, the pigs were euthanized and the nasal mucosae were removed and set in 4% buffered formaldehyde for 24 h at room temperature. Next, the samples were embedded in paraffin blocks and cut into 6 µm sections which were then stained with hematoxylin and eosin before finally viewing them under a microscope (Leica DM2000LED and Leica camera DFC550).

## 3. Results

### 3.1. Solubility Studies

Figure 1 shows the solubilization capacity of DPZ in different oils, surfactants, and cosurfactants. The components that showed the maximum solubilizing potential were selected for the formulation of DPZ-NE. Capryol 90 was used as oil phase, Labrasol as surfactant, and Transcutol-P as cosurfactant.

### 3.2. Pseudo-Ternary Diagrams and Formulation of DPZ-NE and DPZ-PNE

Figure 2 shows the phase diagrams obtained from three different ratios (1:1, 2:1 and 3:1) of Labrasol:Transcutol-P (S_mix_). The phase diagram at S_mix_ ratio of 1:1 exhibited the highest area of emulsification and therefore was selected for the preparation of DPZ-loaded formulation. 

Table 1 details the final formulations of DPZ-NE and DPZ-PNE (6.25 mg/mL). DPZ-NE was prepared by incorporating DPZ in: 6% Capryol 90, 20% Labrasol, 20% Transcutol-P, and 54% water. It was transparent, monophasic, and did not show signs of precipitated drug. DPZ-PNE formed with 24% Pluronic F-127 showed homogeneous and transparent appearance without thermosensitive properties, likely due to the high content of oil, surfactant, and cosurfactant.

### 3.3. Characterization of DPZ-NE and DPZ-PNE

The pH values of DPZ-NE and DPZ-PNE were 5.82 and 6.14, respectively. These values are within the pH range (5.0–6.5) required for nasal formulations [22]. After 24 h of preparation, DPZ-NE showed a mean droplet size around 128.50 ± 1.03 nm with a PI value of 0.12 ± 0.01, which indicated homogeneity of the system. TEM photomicrograph showed small spherical droplets and uniform distribution, which were consistent with the DLS results (Figure 3). The EE (%) of DPZ-NE and DPZ-PNE were 94.32 ± 0.12% and 93.85 ± 0.095%, respectively. These results show a high incorporation of the drug in the inner phase.

Figure 4A,B represents the flow curves τ = f(𝛾̇) and the viscosity curves η = f(𝛾̇) obtained from the rheological characterization of DPZ-NE and DPZ-PNE at 24 h after preparation at 25 °C. The flow curve represented by the relationship between shear stress and shear rate was linear in both formulations.

Table 2 displays the corresponding results of the analysis of the obtained data. DPZ-NE showed nearly constant viscosity values with an increasing shear rate from 0 to 50 s^−1^, which is clearly indicative of Newtonian behavior confirmed by Newton equation fitting. The viscosity of DPZ-NE was about 10.69 ± 0.04 mPa·s. For DPZ-PNE, it was observed that the Ostwald-de Waele model provided the best overall match statistically of the experimental data, which indicates a shear thinning (pseudoplastic) behavior since the viscosity tends to decrease as the shear rate increases. Additionally, the results indicated that the DPZ-PNE viscosity at 50 s^−1^ was 315.40 ± 0.22 mPa·s.

From the ex vivo mucoadhesion study, it was observed that DPZ-PNE exhibited the highest mucoadhesion value of 82.43 ± 1.72% compared with 71.31 ± 1.53% obtained from DPZ-NE. These results indicate the adequacy of DPZ-PNE to adhere to the nasal mucosa, which prolongs the residence time at the site of absorption.

### 3.4. Stability Studies

Figure 5 shows the transmission profiles (%) obtained for DPZ-NE and DPZ-PNE at 4, 25, and 40 °C over a span of 45 days. Peaks on the left and right side of the curve are the result of the meniscus formed by contact between the formulation and the glass. Both DPZ-NE and DPZ-PNE exhibited physical stability without signs of precipitation or changes in the system over 45 days under the conditions studied.

### 3.5. In Vitro Release Study

Figure 6 shows the amount of DPZ released from formulations over time. After 30 h of assay, amounts of 850 and 635.5 µg of DPZ were released from the NE and PNE, respectively. The mathematical equation that best fit experimental data based on the highest coefficient of determination (*r*^2^) was the hyperbolic model for both formulations with *r*^2^ = 0.9283 for DPZ-NE and *r*^2^ = 0.9253 for DPZ-PNE. *p* < 0.05 between B*_max_* (the maximum concentration of drug released) of DPZ-NE (911.1 µg) with respect to B*_max_* of DPZ-PNE (726.3 µg) was observed. The release constant (*K_d_*) was found to be 3.77 h for DPZ-NE and 3.57 h for DPZ-PNE without significant statistical differences between both formulations.

### 3.6. Ex Vivo Permeation Studies

Ex vivo permeation profile of both formulations (Figure 7) shows that the amount of drug permeated through nasal mucosa after 6 h of assay was higher for DPZ-PNE (532.30 µg) compared to DPZ-NE (199.56 µg). 

Table 3 exhibits different permeation and prediction parameters of both formulations. These biopharmaceutical analyses revealed significant statistical differences between DPZ-PNE with respect to DPZ-NE in each one of parameters studied. *J_ss_*, *K_p_*, and *C_ss_* calculated for DPZ-PNE showed values were more than double than those which corresponded to DPZ-NE. With respect to *Q_ret_*, a higher amount of drug retained in nasal mucosa was observed for DPZ-PNE (295.50 µg DPZ/g tissue/cm^2^) when compared with DPZ-NE (192.65 µg DPZ/g tissue/cm^2^).

### 3.7. Cytotoxicity Assay

Figure 8 shows the results of cytotoxicity studies using human nasal cell line RPMI 2650. In both cases, DPZ-NE and DPZ-PNE apparently have dose-dependent cytotoxicity. Cell viability greater than 80% was observed in the assayed dilutions from 3.125 to 25 µg/mL for DPZ-NE and from 3.125 to 12.5 µg/mL for DPZ-PNE. 

### 3.8. Histological Analysis

Histologically, nasal mucosa of negative control (Figure 9A) consisted of a normal mucosa with normal lamina propria with the presence of venules. Nasal mucosa of positive control (Figure 9B) showed infiltration with inflammatory cells and erythrocytes (indicated with an arrow) as well as alteration of the lamina propria. However, the groups treated intranasally with DPZ-NE and DPZ-PNE showed a similar pattern to the negative control without detection of inflammatory signs (Figure 9C,D).

## 4. Discussion

Currently, the treatment of AD is focused on controlling the symptoms by reducing the cholinergic deficiency that is characteristic of the disease using AChE inhibitors to compensate the deficiency of acetylcholine (ACh) in the central nervous system (CNS) [30]. DPZ is widely used in tablet form for oral administration due to its substantial efficacy in improving cognitive functions. Pharmacological efficacy of this drug can be enhanced by the use of alternative routes of administration and development of nanotechnology-based drug delivery systems to overcome the remarkable disadvantages of oral route administration in addition to increasing the bioavailability of the drug in the target area [31]. The connection between the nasal cavity and the brain provides the possibility to bypass the BBB and reach the CNS. Therefore, intranasal administration represents a promising strategy to effectively treat AD [32]. In this study, the development of a NE and PNE was proposed to improve nose-to-brain delivery of DPZ based on their ability to protect the drug from biological and chemical degradation, increase extracellular transport, and provide a high contact surface area. 

DPZ-NE was prepared using the pharmaceutical excipients that exhibited the highest solubilizing potential for the drug (Figure 1). Solubilizing capacity of the oil phase is critical to decrease the proportion of oil used and consequently reduce the amount of surfactant. In this study, Capryol 90 was selected as the oil phase based on the solubility results, Labrasol with a hydrophilic lipophilic balance (HLB) of 14 was used as nonionic O/W surfactant, and Transcutol-P was selected as the cosurfactant based on its solubilizing potential, non-toxic properties, and high biocompatibility [33]. Pseudo-ternary diagram with a mixture of surfactant-cosurfactant (S_mix_) in the weight ratio of 1:1 displayed the greatest NE region and therefore was used for the incorporation of DPZ (Figure 2). DPZ-NE (6.25 mg/mL) was prepared by phase titration method, where spontaneous emulsification is produced by a low energy process [34]. In order to overcome rapid mucociliary clearance of the drug from the nose and increase the mucopenetrative ability of the drug, a non-ionic triblock copolymer of poly(ethylene glycol)-poly(propylene oxide)-poly(ethylene glycol) (PEG-PPO-PEG) was incorporated in the aqueous phase of the formulation to obtain DPZ-PNE. Pluronic F-127 was selected based on its amphiphilic nature, high solubilizing capacity, non-toxicity properties, and the ability to interact with hydrophobic surfaces and biological membranes [35,36].

DPZ-NE and DPZ-PNE were monophasic, transparent, and homogenous with pH values suitable for nasal administration, which suggests that the formulations obtained would not cause nasal irritation [22]. TEM photomicrographs revealed that DPZ-NE was constituted by droplets of spherical shape and dispersed uniformly throughout the system with sizes between 85 and 130 nm, all of which were consistent with the DLS results. 

The rheological analysis supported by mathematical modeling confirmed Newtonian-type flow behavior of DPZ-NE with a constant viscosity when the shear rate increased (Figure 4A), which is ideal for nasal spray application [37]. DPZ-PNE exhibited a shear thinning (pseudoplastic) behavior (Figure 4B) with a viscosity almost 30 times greater than the DPZ-NE viscosity, which favors the nasal mucoretention in order to avoid the mucociliary clearance [38]. 

The ex vivo mucoadhesion study showed that DPZ-PNE exhibited improved bioadhesive properties due to the presence of polymer. Pluronic F-127 is a non-ionic water soluble polymer whose mucoadhesive effect could be related to the rheological properties of the formulation and specific interaction of polymer with the mucosa surface because its amphiphilic nature would allow it to interpenetrate into glycoprotein mucin chains and to form entanglements with mucus, which increases both the drug residence time in the nasal cavity and the drug permeation through nasal mucosa [36,39,40]. Although this bioadhesive effect of Pluronic F-127 is disputed due to its rapid dissolution in aqueous media, the interaction of this polymer with the solvents of DPZ-PNE could favor the adhesive traits of the system, while its surfactant properties allow diffusion via mucous in order to reach the epithelia in a simultaneous manner.

The physical stability studies of both formulations did not detect signs of destabilization, such as creaming, sedimentation, flocculation, or coalescence, over a span of 45 days of study at 4, 25, and 40 °C (Figure 5). The high stability that these vehicles offer is likely due to their small size, which favors the generation of only minimal gravitational force between nanodroplets. Therefore, Brownian motion could be enough to provide the necessary stability to prevent sedimentation or creaming [41].

The ability of formulations to release the drug incorporated into the system was evaluated by in vitro release study using an artificial membrane as the rate-limiting step. The drug release kinetic profile estimated by mathematical models provides crucial information on the formulation and its behavior. Consequently, it is used in quality control studies [42]. DPZ-NE and DPZ-PNE showed faster diffusion of the drug during the first 10 h, followed by a sustained release where it was possible to observe significant differences between the systems (Figure 6). The best kinetic fit was described by hyperbolic profile, which is one of the typical kinetic models for nanostructured systems [43]. 

Ex vivo permeation studies provide useful information to predict in vivo behavior of the formulation. In this research, the drug permeation through nasal mucosa after 6 h of assay was greater for DPZ-PNE compared to DPZ-NE with values that represent 28.39% and 10.64% of the drug placed in the donor compartment, respectively (Figure 7). The high permeability potential of both formulations can be attributed to Labrasol and Transcutol-P, due to their solubilizing capacity and permeation enhancing properties [44]. However, it should be noted that the presence of Pluronic F-127 in DPZ-PNE could provide additional advantages, possibly due to its chemical structure consisting of a hydrophobic core of poly(propylene oxide) between two hydrophilic units of poly(ethylene glycol). Consequently, this makes it a useful surfactant that improves the diffusion ability and permeation of the drug through the mucosa [45,46,47]. In accordance with this approach, the permeation and prediction parameters calculated by biopharmaceutical analysis were significantly higher for DPZ-PNE. This formulation presented a flux or permeation rate more than double than that obtained for DPZ-NE. In the same way, the permeability coefficient proportional to the flux showed a value of 2.13 × 10^−3^ cm/min with respect to 1.05 × 10^−3^ cm/min obtained from DPZ-NE, confirming that the incorporation of Pluronic F-127 constitutes a valuable strategy to improve the drug permeation through nasal mucosa. This approach corroborates with the result obtained from *Q_ret_*, where a high amount of drug retained in nasal mucosa was observed using DPZ-PNE. The combination of bioadhesion and penetration enhancing properties of this formulation increase its permanence and retention in the nasal mucosa while facilitating its transport in order to overcome mucociliary clearance and deliver sustained concentrations of drug to the brain [48]. Since plasma clearance is a constant parameter, when comparing two formulations on the same contact surface the one that facilitates penetration by having higher *K_p_* and inflow values will correlate with high *C_ss_* values. In our study, the highest *K_p_* and flow are exhibited by DPZ-PNE and consequently it showed a value of *C_ss_* 2.11 times greater than DPZ-NE, confirming that *K_p_* and flow are proportional parameters to *C_ss_*. This parameter estimates the concentration of drug that could be reached in the blood in stationary equilibrium state. Previous studies in healthy patients have reported maximum plasmatic concentration of 33.26 ± 6.58 ng/mL for oral administration of 10 mg of DPZ, whereas in this study DPZ-NE and DPZ-PNE showed values of 2.7 and 5.7 times greater, respectively [49]. Based on these results and after further in vivo studies to corroborate our findings, these formulations could be used as therapeutic strategies to increase the efficacy or reduce the dose and/or dosage schedule, thus decreasing the adverse effects of conventional delivery methods [50].

The tolerability of both DPZ-NE and DPZ-PNE was evaluated by in vitro and in vivo models. Cytotoxicity assays using human nasal cell line RPMI 2650 to represent real nasal mucosa had been used previously to test formulations of nanoparticles [51,52]. In this study using this model, no apparent cytotoxicity was observed for either DPZ-NE nor DPZ-PNE. These results were confirmed by in vivo model, where histopathological analysis of porcine nasal mucosa showed that there was no infiltration of inflammatory cells nor significant changes with respect to the negative control after spray application of both DPZ-NE and DPZ-PNE. 

In conclusion, the present research suggests that optimized DPZ-NE and DPZ-PNE could be promising systems for nose-to-brain delivery of DPZ. The comparison of both formulations provides evidence that the incorporation of Pluronic F-127 can be proposed as a reasonable strategy to provide bioadhesion and penetration enhancement in order to prolong residence time at the site of absorption while increasing the penetration of drug through nasal mucosa, thus improving its biopharmaceutical profile. Based on the results of this investigation, further studies are encouraged to use in vivo models to explore the use of these systems as alternatives to improve the efficacy of DPZ or reduce the dose and/or dosage schedule, thus decreasing adverse effects. Furthermore, these systems can be convenient as therapeutic tools in the clinical practice for AD treatment, especially for patients in advanced stages of the disease who could be resistant to cooperation and who might have difficulty swallowing conventional dosage forms. 

## Figures and Tables

**Figure 1 pharmaceutics-11-00064-f001:**
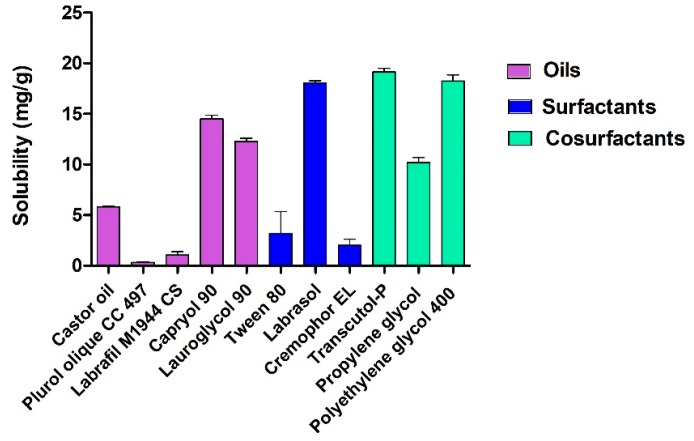
Solubility study of Donepezil (DPZ) in oils, surfactants, and cosurfactants.

**Figure 2 pharmaceutics-11-00064-f002:**
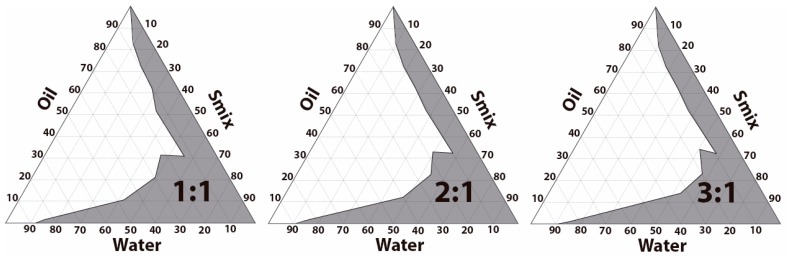
Pseudo-ternary phase diagrams using Capryol 90 as oil phase, Labrasol as surfactant, Transcutol-P as cosurfactant, and water as hydrophilic phase. Labrasol and Transcutol-P were analyzed at different ratios (1:1, 2:1 and 3:1, *w*/*w*).

**Figure 3 pharmaceutics-11-00064-f003:**
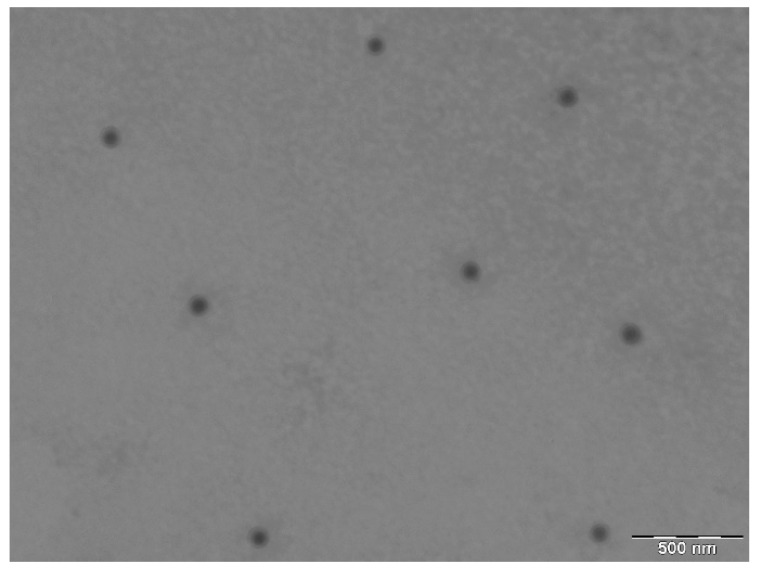
Transmission electron microscopy (TEM) image of DPZ-NE. Magnification 40,000×.

**Figure 4 pharmaceutics-11-00064-f004:**
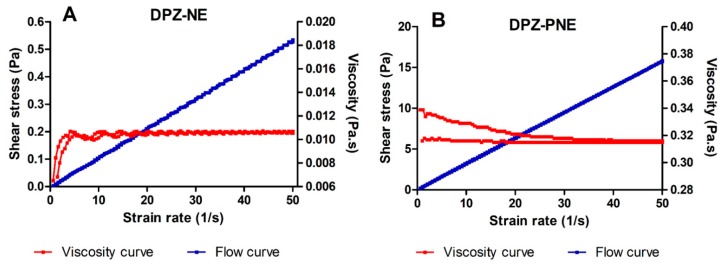
Rheogram showing both flow and viscosity curves. (**A**) DPZ-NE and (**B**) DPZ-PNE.

**Figure 5 pharmaceutics-11-00064-f005:**
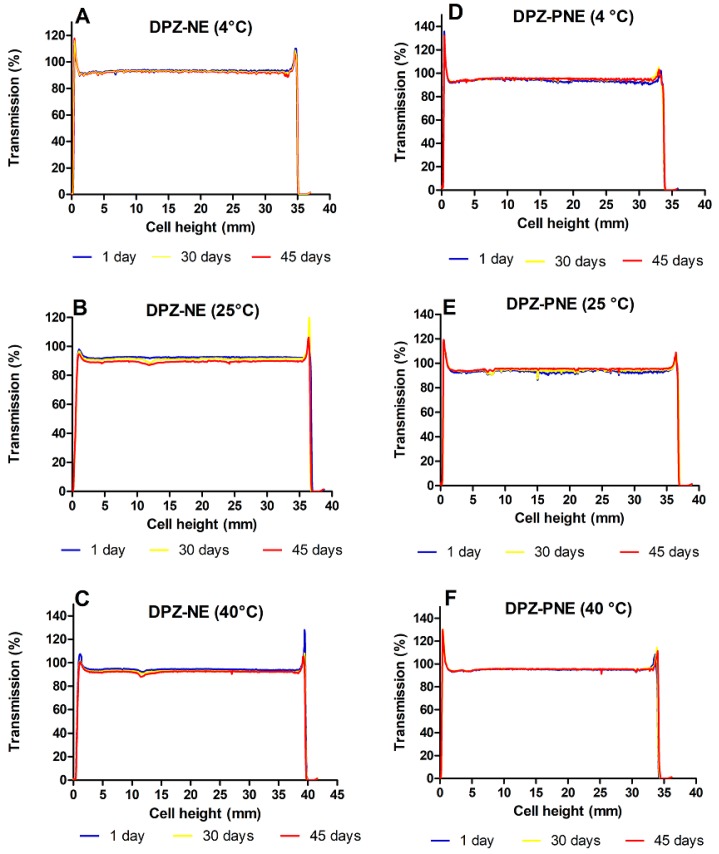
Transmission profiles after 1, 30, and 45 days of production. (**A**) DPZ-NE (4 °C); (**B**) DPZ-NE (25 °C); (**C**) DPZ-NE (40 °C); (**D**) DPZ-PNE (4 °C); (**E**) DPZ-PNE (25 °C), and (**F**) DPZ-PNE (40 °C).

**Figure 6 pharmaceutics-11-00064-f006:**
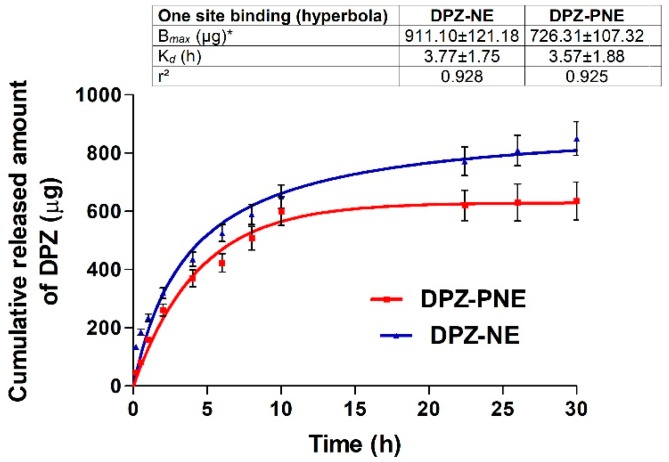
In vitro release profile of DPZ from NE and PNE fitted to Hyperbolic model. Results are expressed as mean ± SD using parametric Student’s *t*-test **p* < 0.05 (*n* = 6).

**Figure 7 pharmaceutics-11-00064-f007:**
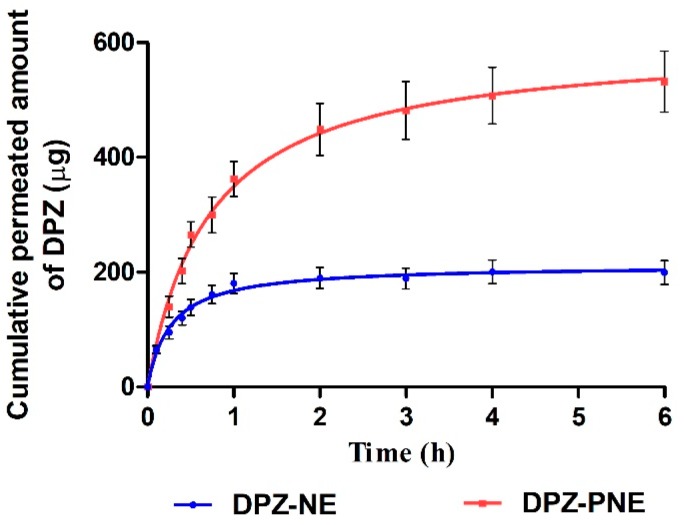
Ex vivo permeation profile of DPZ from NE and PNE through nasal mucosa (*n* = 6).

**Figure 8 pharmaceutics-11-00064-f008:**
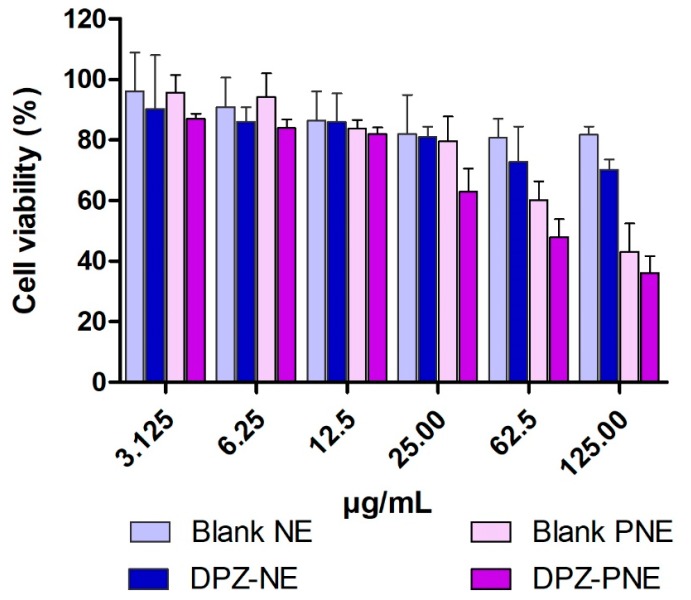
In vitro cytotoxicity studies on human nasal cell line RPMI 2650 of DPZ-NE and DPZ-PNE. The cell viability is recorded as a percentage in contrast to non-treated cells. Results are represented as mean ± SD from 4 independent experiments.

**Figure 9 pharmaceutics-11-00064-f009:**
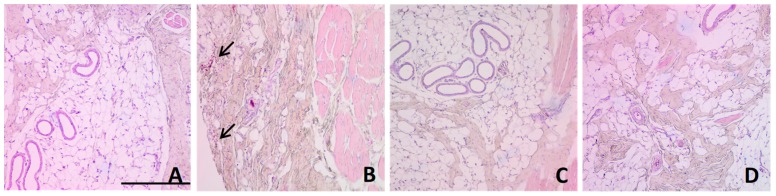
Optical microscopic images of nasal mucosa. (**A**) Negative control (non-treated pig); (**B**) positive control (pig treated with isopropyl alcohol); (**C**) pig treated with DPZ-NE, and (**D**) pig treated with DPZ-PNE. Hematoxylin and eosin stains nuclei blue/black while keratin and cytoplasm are stained red. The arrow indicates infiltration of inflammatory cells. Scale bar = 100 µM.

**Table 1 pharmaceutics-11-00064-t001:** Final formulations of Donepezil-loaded Nanoemulsion (DPZ-NE) and Donepezil-loaded Pluronic F-127 Nanoemulsion (DPZ-PNE).

Components (%)	DPZ-NE	DPZ-PNE
DPZ (6.25 mg/mL)	-	-
Capryol 90	6	6
Labrasol	20	20
Transcutol-P	20	20
Water	54	30
Pluronic F-127	-	24

**Table 2 pharmaceutics-11-00064-t002:** Rotational testing results for DPZ-NE and DPZ-PNE at 25 °C.

Rotational Testing		DPZ-NE	DPZ-PNE
Better mathematical model for fitting	Ramp-up section	Newton *r* = 0.9998	Ostwald de Waele *r* = 1
	Ramp-down section	Newton *r* = 0.9998	Ostwald de Waele *r* = 1
Rheological behavior		Newtonian	Pseudoplastic
Viscosity mean values		10.69 ± 0.04 mPa·s	315.40 ± 0.22 mPa·s

**Table 3 pharmaceutics-11-00064-t003:** Permeation and prediction parameters of DPZ-NE and DPZ-PNE through nasal mucosa.

Formulations	*J_ss_* (µg/(min/cm^2^))	*K_p_* (cm/min) 10^3^	*Q_ret_* (µg DPZ/g tissue/cm^2^)	*C_ss_* (µg/mL)
**DPZ-NE**	6.58 (5.22–7.83)	1.05 (0.83–1.25)	192.65 (108.44–266.26)	0.09 (0.07–0.11)
**DPZ-PNE**	13.30 (12.31–14.07) **	2.13 (1.97–2.25) **	295.50 (239.71–523.36) *	0.19 (0.18–0.21) **

Data are compared for each parameter of DPZ-PNE vs. DPZ-NE * *p* < 0.05, ** *p* < 0.01 by non-parametric Student’s *t*-test (*n* = 6).

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
