# Peer review of "Formulation Strategies to Improve Nose-to-Brain Delivery of Donepezil"

_pharmaceutics, 2019, doi:10.3390/pharmaceutics11020064_

Reviewer 1 Report

The paper deals with an interesting topic such as the alternative administration of donepezil for nose to brain delivery in Alzheimer's patients.

The approach is interesting even if mucoadhesive nanoemulsions have been already been proposed for nose-to-brain delivery in a number of papers and the approach is not completely new.

The authors should concentrate in what is the novelty of the papers compared to existing publications.

In addition some points of the discussion have to be reconsidered. For example a material can not be claimed to be mucoadhesive AND mucopenetrating at the same time. If interaction with mucin is present it is not expected that the material will penetrate inside the mucus layer. In addition, no demonstration of mucus penetrating properties is presented in the paper, while mucoadhesion is experimentally studied. One solution could be, as stated towards the end of the manuscript, a combination of bioadhesion and penetration enhancing properties.

Some of the experimental choices are questionable and should be justified and the details of the criticism are reported here below.

Lines 95-96. The chemical names of Gattefossè products should be reported at least here in the materials section.

Line 189. The choice of a dissolution medium containing methanol and transcutol should need a little discussion since it is not aqueous and its relevance for nasal delivery is questionable. Results obtained with the proposed method are not expected to be representative of an in vivo behaviour.

Line 199. Eliminate the word "skin" from the subsection title.

Line 212. Substitute 0.3 mL to 0.3 L.

Hystological analisys section should report the date and number of the ethic committee approval for in vivo experiments along with volume and method of administration of the formulations.

Line 269. It appears useless to refer to formulations with 18 and 20% plutonic, since not data is actually presented regarding them. Please eliminate any reference to those formulations.

Figure 3. Different scale bars are present on the two panels, hence the magnification can not be 40,000X for both. Please present images with the same magnification. Furthermore images are far from convincing since this is not the typical appearance of stained emulsions (see

Klang V, Matsko NB, Valenta C, Hofer F. Electron microscopy of nanoemulsions: 

an essential tool for characterisation and stability assessment. Micron. 2012

Feb;43(2-3):85-103. doi: 10.1016/j.micron.2011.07.014).

Lines 307-309 and Figure 5. It is not clear at all what is the relevance of the extensibility evaluation for nasal formulations. The experiments has maybe some significance for emulsions for skin applications but none for nasal delivery. Please eliminate from the manuscript.

Table 3 it is not clear how Css is calculated, this should be better explained in the methods and supported in results and discussion sections.

Discussion section need to avoid the repetition of data. Please revise thoroughly the discussion and eliminate references to Figures and to numerical data already presented in Results section.

Lines 387 and 390. The function of surfactants and co-surfactants is basic pharmaceutical science knowledge and should not be included. Eliminate the sentence

Line 428-430. The correlation between the release data and the pharmacodynamics effects is in the present case not possible since the medium selected is not physiologically relevant and the model does not have any significance for in vivo behaviour. Eliminate the sentence.

Lines 434-436. These lines should be moved to results where a better explanation of the model and model parameters is much needed. Avoid the acronym SSD.

Line 441. It seems that the two percentage values have been mentioned in the wrong order and the text should be "28.39% and 10.64%..., respectively".

Line 445. The mucus penetrating effect claimed here is not substantiated since also Labrasol is a polyoxylated tensioattive. The balance between mucoadhesion and mucus penetrating properties require a fine tuning of the properties of the system and here it appears instead that the increase in viscosity and the eventual penetration enhancing properties of pluronic could be at the base of the observed differences. A more relevant discussion should be provided, avoiding oversimplifications.

 Lines 447 and 453. Eliminate reference to "mucus penetration properties" because those properties have not been studied in this manuscript and are just a speculation.

Lines 449-450. Express permeability coefficient as x.xx 10^3 cm/min

Line 474. Substitute "effective" with "promising". The system has not been tested in any in vivo model and its effectiveness is unproven.

Please find a number of text corrections suggested in the edited PDF attached.

Author Response

Reviewer 1:

Questions

The paper deals with an interesting topic such as the alternative administration of donepezil for nose to brain delivery in Alzheimer's patients. The approach is interesting even if mucoadhesive nanoemulsions have been already been proposed for nose-to-brain delivery in a number of papers and the approach is not completely new. The authors should concentrate in what is the novelty of the papers compared to existing publications.

In addition some points of the discussion have to be reconsidered. For example a material can not be claimed to be mucoadhesive AND mucopenetrating at the same time. If interaction with mucin is present it is not expected that the material will penetrate inside the mucus layer. In addition, no demonstration of mucus penetrating properties is presented in the paper, while mucoadhesion is experimentally studied. One solution could be, as stated towards the end of the manuscript, a combination of bioadhesion and penetration enhancing properties.

Answer: In this study, we evaluated the mucoadhesion of both formulations (DPZ-PNE; DPZ-NE). It was observed that the formulation with Pluronic-F127 exhibited greater retention on the nasal mucosa after being washed with PBS. Although this bioadhesive effect of Pluronic-F127 is disputed due to its rapid dissolution in aqueous media, the interaction of this polymer with the solvents of our formulation could favor the adhesive traits of the system while its surfactant properties allow diffusion via mucous in order to reach the epithelia in a simultaneous manner.

Moreover, previous studies that used corneal epithelium have suggested that the high retention using Pluronic-F127 is the result of both rheological properties and specific mucoadhesive interactions, although the exact mechanism is unclear (Al Khateb et al., 2016). With the objective of improving our article and having a stronger argument for the validity of our results, we have incorporated your suggestions with regards to the formulation with Pluronic-F127 as a system with a combination of improved bioadhesive and penetration enhancing properties.

Reference:

Al Khateb, K.; Ozhmukhametova, E. K.; Mussin, M. N.; Seilkhanov, S. K.; Rakhypbekov, T. K.; Lau, W. M.; Khutoryanskiy, V. V., In situ gelling systems based on Pluronic F127/Pluronic F68 formulations for ocular drug delivery. Int J Pharm. 2016, 502, 70-9.

Some of the experimental choices are questionable and should be justified and the details of the criticism are reported here below.

Lines 95-96. The chemical names of Gattefossè products should be reported at least here in the materials section.

Answer: Thank you for your valuable observation. The chemical names of the Gattefossé products that were utilized were reported in the section Materials and Methods in the new manuscript.

Line 189. The choice of a dissolution medium containing methanol and transcutol should need a little discussion since it is not aqueous and its relevance for nasal delivery is questionable. Results obtained with the proposed method are not expected to be representative of an in vivo behaviour.

Answer: A previous study was conducted with different mediums in order to select which would accomplish sink conditions. The medium methanol:transcutol was found to achieve this condition. What’s more is that from a technological standpoint, the in vitro release study is currently used for quality control parameters in order to comply with specifications that are required for manufacturing processes. The objective of the study was to analyze the diffusion kinetic profile of Donepezil from NE in the artificial membrane that showed ideal resistance (Ng et al., 2010). For the in vivo study, this medium will not be employed, whereas for the ex vivo study we used water:trancutol as a medium due to its biocompatibility with the tissue.

Reference:

Shiow-Fern Ng, Jennifer Rouse, Dominic Sanderson, Gillian Eccleston. Pharmaceutics. 2010, 2(2),209-223; doi:10.3390/pharmaceutics2020209

Line 199. Eliminate the word "skin" from the subsection title.

Answer: There had been an untimely error in the title of this subsection but it has since been rectified in the new manuscript thanks to your astute observation.

Line 212. Substitute 0.3 mL to 0.3 L.

Answer: The manuscript was modified to reflect the proper measurement used during experiments.

Hystological analisys section should report the date and number of the ethic committee approval for in vivo experiments along with volume and method of administration of the formulations.

Answer: This assay was performed in animals that were subjected to surgeries for medical studies in the Animal Facility at the Bellvitge Campus of the University of Barcelona in accordance with both the Animal Experimentation Ethical Committee of the University of Barcelona, Spain (CEEA-UB) and Animal Experimentation Commission of the Generalitat de Catalunya.   Before the surgical procedure and while under deep anesthesia, a volume of 300 µL of developed formulation or isopropyl alcohol (positive control) was administered using a nasal spray pump. After sacrifice, the biological samples were donated for histological analysis in order to apply the concept of 3R. Additionally, we have attached a letter that explains the aforementioned argument.

Line 269. It appears useless to refer to formulations with 18 and 20% plutonic, since not data is actually presented regarding them. Please eliminate any reference to those formulations.

Answer: All references to formulations with 18 and 20% Plutonic have subsequently been eliminated.

Figure 3. Different scale bars are present on the two panels, hence the magnification can not be 40,000X for both. Please present images with the same magnification. Furthermore images are far from convincing since this is not the typical appearance of stained emulsions (see Klang V, Matsko NB, Valenta C, Hofer F. Electron microscopy of nanoemulsions: an essential tool for characterisation and stability assessment. Micron. 2012 Feb;43(2-3):85-103. doi: 10.1016/j.micron.2011.07.014).

Answer: I appreciate this valuable observation. As you mentioned, the magnification used for both formulations were not the same. There was, in fact, an error that we did not detect while writing and reviewing the manuscript. Indeed, the magnification of 40,000X corresponds to the NE, whereas it was not possible to visualize DPZ-PNE using the same magnification. For this reason, we decided to remove the photographs that correspond to DPZ-PNE.

With respect to the appearance of our NE, we observed dark droplets of spherical shape and a size in accordance with our DLS results as well as other studies in which the same oil phase (Capryol 90) was used. Of these, a similar shape of the droplets was obtained (Yang et al., 2017; Azeem et al., 2009). We also reviewed the suggested article which provides invaluable information pertaining to the evaluation of the morphology of the NE. It was evident that the cryo techniques of sample preparation for both TEM and SEM provide better quality images and will therefore be considered for future studies.

References:

Yang M, Gu Y, Yang D, Tang X and Liu J. Development of triptolide-nanoemulsion gels for percutaneous administration: physicochemical, transport, pharmacokinetic and pharmacodynamic characteristics. J Nanobiotechnol (2017) 15:88.

Azeem, A.; Rizwan, M.; Ahmad, F. J.; Iqbal, Z.; Khar, R. K.; Aqil, M.; Talegaonkar, S., Nanoemulsion components screening and selection: a technical note. AAPS PharmSciTech. (2009), 10, 69-76.

Lines 307-309 and Figure 5. It is not clear at all what is the relevance of the extensibility evaluation for nasal formulations. The experiments has maybe some significance for emulsions for skin applications but none for nasal delivery. Please eliminate from the manuscript.

Answer: All references to extensibility evaluation for nasal formulations were removed, as per your recommendation.

Table 3 it is not clear how Css is calculated, this should be better explained in the methods and supported in results and discussion sections.

Answer: Once more, we are grateful for your guidance. A detailed description about the determination of Css in addition to the interpretation of its result were incorporated to the corresponding sections of the new manuscript.

Discussion section need to avoid the repetition of data. Please revise thoroughly the discussion and eliminate references to Figures and to numerical data already presented in Results section.

Answer: The discussion was thoroughly revised and edited in order to minimize redundancy while, at the same, maintaining clarity.

Lines 387 and 390. The function of surfactants and co-surfactants is basic pharmaceutical science knowledge and should not be included. Eliminate the sentence

Answer: The aforementioned sentence was eliminated.

Line 428-430. The correlation between the release data and the pharmacodynamics effects is in the present case not possible since the medium selected is not physiologically relevant and the model does not have any significance for in vivo behaviour. Eliminate the sentence.

Answer: The aforementioned sentence was edited in the text as follows:

The drug release kinetic profile estimated by mathematical models provides crucial information on the formulation and its behavior (Balzus et al., 2016).

Reference:

Balzus, B.; Colombo, M.; Sahle, F. F.; Zoubari, G.; Staufenbiel, S.; Bodmeier, R., Comparison of different in vitro release methods used to investigate nanocarriers intended for dermal application. Int J Pharm. 2016, 513, 247-254; 10.1016/j.ijpharm.2016.09.033.

Lines 434-436. These lines should be moved to results where a better explanation of the model and model parameters is much needed. Avoid the acronym SSD.

Answer: These lines were placed in the section that you suggested. Indeed, they help clarify the explanation of the model and model parameters.

Line 441. It seems that the two percentage values have been mentioned in the wrong order and the text should be "28.39% and 10.64%..., respectively".

Answer. As a matter of fact, this was an error and your detection of it is greatly appreciated -- thank you for your attention to detail.

Line 445. The mucus penetrating effect claimed here is not substantiated since also Labrasol is a polyoxylated tensioattive. The balance between mucoadhesion and mucus penetrating properties require a fine tuning of the properties of the system and here it appears instead that the increase in viscosity and the eventual penetration enhancing properties of pluronic could be at the base of the observed differences. A more relevant discussion should be provided, avoiding oversimplifications.

Answer: The new manuscript includes a more in-depth and detailed discussion with respect to the role of Pluronic in the bioadhesion and permeation of the drug. The added paragraph is as follows:

In this research, the drug permeation through nasal mucosa after 6 h of assay was greater for DPZ-PNE compared to DPZ-NE with values that represent 28.39 and 10.64% of the drug placed in the donor compartment, respectively. The high permeability potential of both formulations can be attributed to Labrasol and Transcutol-P due to their solubilizing capacity and permeation-enhancing properties. However, it should be noted that the presence of Pluronic-F127 in DPZ-PNE could provide an additional advantage in drug permeation, possibly due to its chemical structure consisting of a hydrophobic core of poly(propylene oxide) between two hydrophilic units of poly(ethylene glycol). Consequently, this makes it a useful surfactant that improves the diffusion ability and permeation of the drug through the mucosa (Giuliano et al., 2018; Lai et al., 2009; Kim et al., 2011).

References:

Giuliano, E.; Paolino, D.; Fresta, M.; Cosco, D., Mucosal Applications of Poloxamer 407-Based Hydrogels: An Overview. Pharmaceutics. 2018, 10.

Lai, S. K.; Wang, Y. Y.; Hanes, J., Mucus-penetrating nanoparticles for drug and gene delivery to mucosal tissues. Adv Drug Deliv Rev. 2009, 61, 158-71.

Kim, K. A.; Lim, J. L.; Kim, C.; Park, J. Y., Pharmacokinetic comparison of orally disintegrating and conventional donepezil formulations in healthy Korean male subjects: a single-dose, randomized, open-label, 2-sequence, 2-period crossover study. Clin Ther. 2011, 33, 965-72.

 Lines 447 and 453. Eliminate reference to "mucus penetration properties" because those properties have not been studied in this manuscript and are just a speculation.

Answer: The suggested modification was implemented in order to avoid any misunderstandings.

Lines 449-450. Express permeability coefficient as x.xx 10^3 cm/min

Answer: The unit of permeability coefficient was expressed at the end of the numerical value as per your suggestion.

Line 474. Substitute "effective" with "promising". The system has not been tested in any in vivo model and its effectiveness is unproven.

Answer: Thank you for sharing this valuable observation. The word “effective” was substituted for “promising” in the corresponding section.

Reviewer 2 Report

Comments:

This paper deals with the preparation and characterization of mucoadhesive-nanoemulsions for Donepezil (DPZ) delivery by nasal administration in order to obtain a brain release

- Explain better how Pluronic F127 enhance mucoadhesion.

- The authors must better explain the temperature chosen for mucoadhesive studies (37°C), in fact 32 °C represents the temperature in line with the nasal physiological temperature. A high temperature in fact, could be influence or enhance NE permeation across membrane.

-For the mucoadhesive studies, the authors take in account the presence of the mucus on porcine nasal mucosa? (E.g.Flávia Nathiely Silveira Fachel et al. 2018, Cararbohydrate Polymers, 199,572-582)

-In my opinion stability studies in artificial CSF (cerebro spinal fluid) could be carried out, for 3 h at 37°C, following size variation by DLS to confirm the release by not degraded NE. (E.g. Rinaldi et al. 2019, Journal of Controlled Release,294, 17-26)

- What is the entrapment efficiency of DZP in NE? What is the concentration useful to obtain therapeutic effect (lower than the oral administration dose)? What is the solvent employed to solubilize the drug in order to obtain calibration curve? In Fig 7 add in the graph the axis of DPZ percent release.

- Mucoadhesive experiment with mucin could be carried out following NE+mucin stability by DLS.

-In table 1, the percent must be referred to component: e.g. Labrasol % …20.

Author Response

Reviewer 2:

Questions

This paper deals with the preparation and characterization of mucoadhesive-nanoemulsions for Donepezil (DPZ) delivery by nasal administration in order to obtain a brain release

1.      Explain better how Pluronic F127 enhance mucoadhesion.

Answer: This is a valuable observation that improves the overall text. The following information was added to the manuscript concerning the role of Pluronic F-127 in the mucoadhesion:

The ex vivo mucoadhesion study showed that DPZ-PNE exhibited improved bioadhesive properties due to the presence of polymer. Pluronic F-127 is a non-ionic water soluble polymer whose mucoadhesive effect could be related to the rheological properties of the formulation and specific interaction of polymer with the mucosa surface because its amphiphilic nature would allow it to interpenetrate into glycoprotein mucin chains and to form entanglements with mucus which increases both the drug residence time in the nasal cavity and the drug permeation through nasal mucosa (Al Khateb et al, 2016; Dumortier et al, 2006; Chatterjee et al 2017). Although this bioadhesive effect of Pluronic-F127 is disputed due to its rapid dissolution in aqueous media, the interaction of this polymer with the solvents of our formulation could favor the adhesive traits of the system while its surfactant properties allow diffusion via mucous in order to reach the epithelia in a simultaneous manner.

References:

-          Al Khateb, K.; Ozhmukhametova, E. K.; Mussin, M. N.; Seilkhanov, S. K.; Rakhypbekov, T. K.; Lau, W. M.; Khutoryanskiy, V. V., In situ gelling systems based on Pluronic F127/Pluronic F68 formulations for ocular drug delivery. Int J Pharm. 2016, 502, 70-9; 10.1016/j.ijpharm.2016.02.027.

-          Dumortier, G.; Grossiord, J. L.; Agnely, F.; Chaumeil, J. C., A review of poloxamer 407 pharmaceutical and pharmacological characteristics. Pharm Res. 2006, 23, 2709-28; 10.1007/s11095-006-9104-4.

-          Chatterjee, B.; Amalina, N.; Sengupta, P.; Kumar, U., Mucoadhesive Polymers and Their Mode of Action: A Recent Update. Journal of Applied Pharmaceutical Science. 2017, 7, 195-203.

2.       The authors must better explain the temperature chosen for mucoadhesive studies (37°C), in fact 32 °C represents the temperature in line with the nasal physiological temperature. A high temperature in fact, could be influence or enhance NE permeation across membrane.

Answer: We have performed this assay in accordance with approaches from other authors and, considering that the sample was not kept inside a temperature controlled chamber, we only used PBS previously warmed at 37 °C (Sha et al., 2015; Swamy and Abbas, 2011). However, your observation is indeed valuable and and will therefore be considered for future studies in order to obtain more precise results.

References:

-          Shah, B. M.; Misra, M.; Shishoo, C. J.; Padh, H., Nose to brain microemulsion-based drug delivery system of rivastigmine: formulation and ex-vivo characterization. Drug Deliv. 2015, 22, 918-30.

-          Swamy, N.; Abbas, Z., Preparation and In Vitro Characterization of Mucoadhesive Hydroxypropyl Guar Microspheres Containing Amlodipine Besylate for Nasal Administration. Indian J Pharm Sci. 2011, 73, 608–614.

3.      For the mucoadhesive studies, the authors take in account the presence of the mucus on porcine nasal mucosa? (E.g.Flávia Nathiely Silveira Fachel et al. 2018, Cararbohydrate Polymers, 199,572-582).

Answer: Once more, we are grateful for your guidance. The methodology of this assay was incomplete in the manuscript so your observation is indeed accurate, which has led us to accommodate the addition of this observation. Previously, we hydrated the nasal mucosa with artificial nasal mucus prepared with 8% of mucin from porcine stomach (Sigma Aldrich) dispersed in a solution of 7.45 mg/mL NaCl, 1.29 mg/mL, 1.29 mg/mL KCl and 0.32 mg/mL CaCl2·2H2O (Colombo, et al., 2018).

Reference:

-          Colombo, M.; Figueiro, F.; de Fraga Dias, A.; Teixeira, H. F.; Battastini, A. M. O.; Koester, L. S., Kaempferol-loaded mucoadhesive nanoemulsion for intranasal administration reduces glioma growth in vitro. Int J Pharm. 2018, 543, 214-223; 10.1016/j.ijpharm.2018.03.055

4.      In my opinion stability studies in artificial CSF (cerebro spinal fluid) could be carried out, for 3 h at 37°C, following size variation by DLS to confirm the release by not degraded NE. (E.g. Rinaldi et al. 2019, Journal of Controlled Release,294, 17-26).

Answer: We have carried out the study that you suggested. After mixing DPZ-NE in the CSF at 45%, we obtained an exponential kinetic model showing a destabilization of system at first contact with CFS. This effect could be the result of an intensification of interfacial tension caused by the increase of the proportion of aqueous solution in the system, which would lead to flocculation of the droplets.

For the 90% of DNZ-NE in the fluid, stability was observed in these systems around 60 min. Taking into account that the NE is only a vehicle to improve the delivery of the drug and the intranasal route being a rapid absorption route, it would take less than 60 minutes to for drug action to occur.

5.      What is the entrapment efficiency of DZP in NE?

This parameter was not evaluated previously however it was included in the new manuscript. The result of this assay showed a high incorporation of the drug in the inner phase with values of 94.32±0.12 and 93.85±0.095% for DPZ-NE and DPZ-PNE, respectively.

6.      What is the concentration useful to obtain therapeutic effect (lower than the oral administration dose)?

The concentration for oral administration is 5, 10 and 23 mg DPZ. However, considering the advantages of intranasal route and the low volume of administration, we have used 300 µl of formulation that represent 1.875 mg of DPZ. Biopharmaceutical analysis showed that with this amount assayed, the theoretical human steady-state plasma concentration (Css) for DPZ-PNE and DPZ-NE was 0.19 µg/mL and 0.09 µg/mL, respectively. Compared to other studies, after oral administration of 10 mg of DPZ the maximum plasmatic concentration reported was 33.26±6.58 ng/mL and hence DPZ-NE and DPZ-PNE showed values of 2.7 and 5.7 times greater, which suggest carrying out further in vivo studies to corroborate our findings because it could allow reduction of the dose and/or dosage schedule of the drug.

7.      What is the solvent employed to solubilize the drug in order to obtain calibration curve?

Answer: The calibration curve was performed using methanol as solvent to solubilize DPZ.

8.       In Fig 7 add in the graph the axis of DPZ percent release.

Answer: According to many articles, we have expressed the values in μg since the expression of percentage released does not allow the calculation of the parameters in terms of units that describe the kinetics. Nevertheless, it is interesting to add the total % released of the drug to each formulation at 30 h, being approximately 46% for the DPZ-NE and 34% for the DPZ-PNE (Sanz et al., 2018; Silva-Abreu et al., 2018).

References:

-          Sanz, R.; Clares, B.; Mallandrich, M.; Suner-Carbo, J.; Montes, M. J.; Calpena, A. C., Development of a mucoadhesive delivery system for control release of doxepin with application in vaginal pain relief associated with gynecological surgery. Int J Pharm. 2018, 535, 393-401.

-          Silva-Abreu, M.; Calpena, A. C.; Espina, M.; Silva, A. M.; Gimeno, A.; Egea, M. A.; Garcia, M. L., Optimization, Biopharmaceutical Profile and Therapeutic Efficacy of Pioglitazone-loaded PLGA-PEG Nanospheres as a Novel Strategy for Ocular Inflammatory Disorders. Pharm Res. 2018, 35, 11.

9.       Mucoadhesive experiment with mucin could be carried out following NE+mucin stability by DLS.

Answer: We have carried out the stability studies that you aptly suggested using artificial nasal mucus (ANM) at three different ratios of NE:ANM (1:1, 0.5:1, 0.25:1). Similarly, as with the stability studies in artificial CSF, the results showed a destabilization of the systems proportional to the volume of fluid used. However, the droplet size remained not greater than 200 nm at the ratios of 0.5:1 and 0.25:1.

-In table 1, the percent must be referred to component: e.g. Labrasol % …20.

Answer: Thank you for your observation. We have edited Table 1 in the manuscript to account for this modification.

Round  2

Reviewer 1 Report

The comments were addressed properly

Reviewer 2 Report

The authors reply in appropriate way to my comments.

So I think that this research article could be accepted in the present form.